# Fully Automated Thrombus Segmentation on CT Images of Patients with Acute Ischemic Stroke

**DOI:** 10.3390/diagnostics12030698

**Published:** 2022-03-12

**Authors:** Mahsa Mojtahedi, Manon Kappelhof, Elena Ponomareva, Manon Tolhuisen, Ivo Jansen, Agnetha A. E. Bruggeman, Bruna G. Dutra, Lonneke Yo, Natalie LeCouffe, Jan W. Hoving, Henk van Voorst, Josje Brouwer, Nerea Arrarte Terreros, Praneeta Konduri, Frederick J. A. Meijer, Auke Appelman, Kilian M. Treurniet, Jonathan M. Coutinho, Yvo Roos, Wim van Zwam, Diederik Dippel, Efstratios Gavves, Bart J. Emmer, Charles Majoie, Henk Marquering

**Affiliations:** 1Department of Biomedical Engineering and Physics, Amsterdam UMC, 1105 AZ Amsterdam, The Netherlands; m.l.tolhuisen@amsterdamumc.nl (M.T.); h.vanvoorst@amsterdamumc.nl (H.v.V.); n.arrarteterreros@amsterdamumc.nl (N.A.T.); p.r.konduri@amsterdamumc.nl (P.K.); h.a.marquering@amsterdamumc.nl (H.M.); 2Department of Radiology and Nuclear Medicine, Amsterdam UMC, 1105 AZ Amsterdam, The Netherlands; m.kappelhof@amsterdamumc.nl (M.K.); a.e.bruggeman@amsterdamumc.nl (A.A.E.B.); bruna.gdutra@gmail.com (B.G.D.); j.w.hoving@amsterdamumc.nl (J.W.H.); b.j.emmer@amsterdamumc.nl (B.J.E.); c.b.majoie@amsterdamumc.nl (C.M.); 3Nicolab, 1105 BP Amsterdam, The Netherlands; eponomareva@nico-lab.com (E.P.); ijansen@nico-lab.com (I.J.); 4Department of Radiology, Catharina Ziekenhuis, 5623 EJ Eindhoven, The Netherlands; lonneke.yo@catharinaziekenhuis.nl; 5Department of Neurology, Amsterdam UMC, 1105 AZ Amsterdam, The Netherlands; n.e.lecouffe@amsterdamumc.nl (N.L.); j.brouwer@amsterdamumc.nl (J.B.); j.coutinho@amsterdamumc.nl (J.M.C.); y.b.roos@amsterdamumc.nl (Y.R.); 6Department of Medical Imaging, Radboud UMC, 6525 GA Nijmegen, The Netherlands; anton.meijer@radboudumc.nl; 7Medical Imaging Center, UMC Groningen, 9713 GZ Groningen, The Netherlands; a.p.a.appelman@umcg.nl; 8Research Bureau of Radiology and Nuclear Medicine, Amsterdam UMC, 1105 AZ Amsterdam, The Netherlands; k.m.treurniet@amsterdamumc.nl; 9Department of Radiology, The Hague Medical Center, 2262 BA The Hague, The Netherlands; 10Department of Radiology and Nuclear Medicine, Maastricht UMC, Cardiovascular Research Institute Maastricht (CARIM), 6229 HX Maastricht, The Netherlands; w.van.zwam@mumc.nl; 11Department of Neurology, Erasmus MC UMC, 3015 GD Rotterdam, The Netherlands; d.dippel@erasmusmc.nl; 12Informatics Institute, University of Amsterdam, 1098 XH Amsterdam, The Netherlands; e.gavves@uva.nl

**Keywords:** ischemic stroke, thrombus, segmentation, CT imaging, CT angiography, convolutional neural network (CNN), U-Net

## Abstract

Thrombus imaging characteristics are associated with treatment success and functional outcomes in stroke patients. However, assessing these characteristics based on manual annotations is labor intensive and subject to observer bias. Therefore, we aimed to create an automated pipeline for consistent and fast full thrombus segmentation. We used multi-center, multi-scanner datasets of anterior circulation stroke patients with baseline NCCT and CTA for training (n = 228) and testing (n = 100). We first found the occlusion location using StrokeViewer LVO and created a bounding box around it. Subsequently, we trained dual modality U-Net based convolutional neural networks (CNNs) to segment the thrombus inside this bounding box. We experimented with: (1) U-Net with two input channels for NCCT and CTA, and U-Nets with two encoders where (2) concatenate, (3) add, and (4) weighted-sum operators were used for feature fusion. Furthermore, we proposed a dynamic bounding box algorithm to adjust the bounding box. The dynamic bounding box algorithm reduces the missed cases but does not improve Dice. The two-encoder U-Net with a weighted-sum feature fusion shows the best performance (surface Dice 0.78, Dice 0.62, and 4% missed cases). Final segmentation results have high spatial accuracies and can therefore be used to determine thrombus characteristics and potentially benefit radiologists in clinical practice.

## 1. Introduction

Ischemic stroke treatment outcome is shown to be associated with thrombus imaging characteristics, such as perviousness, length, and distance from the internal carotid artery terminus (ICA-T) [1,2]. In order to calculate thrombus perviousness and length, the location of the thrombus should first be detected. Next, the extent of the thrombus should be annotated or marked by several volumes of interest (VOIs). An alternative to this partial thrombus annotation involves segmenting the entire thrombus, which leads to a more accurate calculation of thrombus imaging characteristics [3]. Furthermore, radiomic features, which are also shown to be associated with treatment outcome [4,5], can only be extracted from full thrombus segmentations. However, manual annotations or segmentations of the entire thrombus are inconsistent and oftentimes inaccurate. Additionally, they are highly time consuming and cannot be done in the acute setting of stroke. Currently, thrombus location is the only thrombus imaging characteristic that is taken into account in ischemic stroke patient management [6]. An automatic thrombus segmentation method, however, can result in accurate and fast determination of other thrombus imaging characteristics, and consequently facilitate their use in stroke patient assessment.

The most common imaging modality for the acute diagnostic work-up in stroke is computed tomography (CT). On non-contrast CT (NCCT) scans, thrombi may appear as a hyper-attenuated vascular area, which is known as the “hyperdense artery sign” (HAS) [7]. The attenuation of an HAS has a positive correlation with the concentration of red blood cells (RBCs) in the thrombus [8]. HAS is one of the earliest signs of ischemic stroke and can be detected with high sensitivity and specificity, which indicates that the majority of patients with confirmed acute ischemic stroke show a visible hyperdensity on NCCT [9]. Additionally, thin slice reconstruction of NCCT improves HAS visibility [10]. Therefore, the detection of HAS is improved as acquisition of NCCT with thinner slices becomes possible.

On CT angiography (CTA), the location of the proximal side of the thrombus can be detected as a termination or disruption of the contrast-enhanced artery. In case of sufficient perfusion of the distal side of the occlusion by collaterals, the distal side can also be correctly inferred. However, insufficient retrograde filling can result in over-estimation of the thrombus size on CTA [11]. Therefore, if HAS is visible on NCCT, it may be a more reliable indicator of the thrombus extent than a lack of attenuation on CTA.

Several clinically validated software solutions exist which provide large vessel occlusion (LVO) detection on NCCT scans [12] or CTA scans, including Brainomix e-Stroke Suite, iSchemaView RAPID [13], Viz. ai LVO [14] and Nicolab StrokeViewer LVO. However, there are only a few studies that address thrombus segmentation on CT scans. The main challenges in the thrombus segmentation task are (1) the small size of the thrombi compared to the entire brain, (2) the occasional existence of other highly attenuated vascular areas on NCCT that mimic HAS, such as calcified arteries, high hematocrit, and beam hardening artefacts [9], (3) lack of HAS on NCCT for thrombi with low concentrations of RBCs, and (4) over-estimation of thrombus on CTA due to insufficient collateral flow.

Thrombus segmentation on CT scans was first reported by Santos et al. [11] who developed a semi-automated method, which uses manually selected proximal and distal seed points and the contralateral artery to segment the thrombus on CTA scans. Beesomboon et al. [15] extract a number of brain regions with similar intensity on NCCT and then proceed to eliminate the non-thrombus regions with a rule-based algorithm. However, they only provide visual comparison with ground truth without reporting any quantifiable measures.

Convolutional neural networks (CNN)s have shown great promise in medical image segmentation. Lucas et al. [16] use two cascaded CNNs to segment thrombi on NCCT. The first network is a U-Net architecture that takes a region of interest (ROI) containing middle cerebral artery (MCA) and ICA regions as input and segments all the candidate regions in it. The second network is used for false positive reduction by predicting whether the left, right or none of the hemispheres contains a thrombus. They achieve a Dice score of 0.5 on a small test set. Tolhuisen et al. [12] train two patch-based CNNs: one for detecting asymmetry between hemispheres and another for HAS detection on NCCT scans. First, the image patches are classified using both networks, and then voxel-wise segmentation is done on the patches that were classified as positive using the HAS detection network. However, even though their method is able to detect thrombi with the same accuracy as expert neuroradiologists, the volumetric, and spatial agreement of the segmentation predictions with the ground truth is low.

In this study, we present and validate a proof-of-concept for automatic thrombus segmentation using dual-modality U-Net based CNNs. We use an off-the-shelf algorithm, StrokeViewer LVO (Nicolab, Amsterdam, The Netherlands; www.nicolab.com/strokeviewer-home accessed on 12 March 2022), to detect the thrombus location and then limit the search area by creating a bounding box around the detected thrombus location. The overlapping NCCT and CTA bounding boxes are subsequently given as input to the CNNs to automatically segment the thrombi. We introduce the dynamic bounding box algorithm to improve the chance of enclosing the entire thrombus inside the final bounding box. Limiting the segmentation space alleviates the problem of size disparity between thrombus and brain. Additionally, it simplifies the segmentation task by omitting some of the potential HAS mimics. Furthermore, despite NCCT being a more reliable information source for thrombus segmentation, it is easier to detect the thrombus based on the vessels shown on CTA. Both modalities can therefore be more effectively utilized when the detection and segmentation steps are separated.

## 2. Materials and Methods

### 2.1. Dataset

Baseline NCCT and CTA scans of 228 patients with an anterior cerebral large vessel occlusion (LVO) stroke (ICA, M1, and M2 segment of the MCA) from two datasets were included for the train and validation set. The first dataset contains 129 patients from the multicenter randomized clinical trial of endovascular treatment for acute ischemic stroke in The Netherlands (MR CLEAN) [17]. The second dataset contains fully anonymized data from a globally collected cohort of 99 patients with confirmed large vessel occlusions in the anterior cerebral circulation. The data were collected according to local standards and regulations. The train set contains 208 scans, 115 of which are from the first dataset. The validation set contains 20 scans, 14 of which are from the first dataset. The scans are randomly divided between the train and validation set.

The test set consists of baseline NCCT and CTA scans of 100 randomly chosen patients from the MR CLEAN-NO IV trial [18].

Patients with an LVO of the anterior circulation from twenty centers in The Netherlands, France, and Belgium were included in this trial between 2018 and 2020. Patients were excluded for the following reasons: NCCT scans with slice thickness ≥5 mm (n = 29), no visible HAS at all which makes accurate annotation of thrombus impossible (n = 7), CTA slice thickness >2 mm or minimum number < 8, incomplete brain, or sharp convolutional kernels (such as “boneplus”, “h70h”, “FC30”, or “FC31”) (n = 18), failed automatic registration (n = 4), patients with extracranial occlusions (n = 3), and no occlusion detected by StrokeViewer (n = 15).

The 15 cases where StrokeViewer LVO did not produce a prediction include proximal and distal M2 (n = 11), ICA-T (n = 2), and M1 (n = 2) occlusions. The 100 test set scans include ICA-T (n = 22), M1 (n = 70) and proximal and distal M2 (n = 8) occlusions.

Descriptive characteristics of the thrombi in the training, validation, and test set are summarized in Table 1.

#### 2.1.1. Ground Truth: Manual Annotations

All of the thrombus delineations were initially made by a trained observer (MM) based on both NCCT and CTA scans. The annotations for the scans in training and validation set were checked by a trained observer with 10 years of annotating experience (IJ). The annotations for the scans in the test set were checked by a neuroradiologist with over 15 years of experience in neuroradiology (BE).

#### 2.1.2. Pre-Processing

The pre-processing steps included skull stripping, registering the NCCT and CTA scans to the MNI-305 atlas brain [19], clipping the intensity values in the NCCT and CTA images between (0, 100) and (0, 250) HU, respectively, and normalizing the resulting intensity values between (0, 1). Skull stripping was performed using a deep learning based method developed by Nicolab. For registration, the affine registration feature of SimpleElastix software [20] was used.

### 2.2. Modelling and Analysis

Our method first acquired the location of the thrombus using LVO detection and localization of StrokeViewer. Subsequently, an initial bounding box was created around the thrombus location on both CTA and NCCT. Different CNNs were used to segment the thrombus inside the bounding box. Our proposed dynamic bounding box algorithm adjusted the size and location of the bounding box to improve the chance of the entire thrombus being segmented by the network. The algorithms used in this study are available upon reasonable request.

#### 2.2.1. Segmentation Networks

Since both NCCT and CTA contain important information about the thrombus, we used multi-modal networks for thrombus segmentation. All segmentation networks were based on a fully convolutional encoder–decoder architecture similar to U-Net. We evaluated four ways of combining the two CT modalities (NCCT and CTA) in the U-Net: (1) one was to give the two modalities as two input channels to the U-Net. Another option was to have a separate encoder for each modality and combine the features by (2) concatenation, (3) adding, or (4) weighted sum. Additionally, we report the results of using only NCCT as the single input modality to the U-Net network as a baseline to examine the effect of including information from CTA.

Since the dynamic bounding box algorithm creates bounding boxes of different sizes, our networks should be capable of accepting inputs of various sizes. Although U-Net is fully convolutional, the existence of skip connections between the encoder and the decoder results in a limited set of acceptable input sizes for this network. Therefore, we transformed each bounding box to the closest acceptable input size by zero padding. The models were implemented using PyTorch 1.9.0 [21].

##### U-Net

In the U-Net architecture, we concatenated the NCCT and CTA images in the beginning and fed them as a single matrix with two channels to the U-Net. Consequently, the features were combined in the first convolutional layer of the network. The details of this network are depicted in Figure 1. All 3D convolutional layers had a kernel size of 3 and a padding size of 1. The kernel size for both MaxPool and UpConv operations was 2. The upsampling algorithm used for UpConv was ‘trialinear’ with align corners set to “True”. The number of channels for each layer are depicted in Figure 1.

##### Concatenate

In the network with the concatenate feature fusion, we combined the features at a deeper level by using a separate encoder for each modality. Having separate encoder paths allows the network to create more specialized features for each modality. The architecture of this network is depicted in Figure 2. The final features of both encoders were combined and given as input to the decoder. Additionally, the features of the two encoders were combined at each level and passed to the decoder by residual connections. All of the feature fusions were done by concatenation. Feature fusion by concatenation is also used in the skip connections in U-Net, DenseNet [22], and the inception block in GoogLeNet [23].

##### Add

Combining the features with concatenation is a common feature fusion method. However, it significantly increases the number of parameters in the model. An alternative common feature fusion method is to add the two feature matrices together. With this method, no additional trainable parameters are required for the fusion process. Feature fusion by addition is also used in the residual connections in ResNet [24] and the feature pyramid network (FPN) [25].

##### Weighted-Sum

In addition to the two common feature fusion methods of concatenation and adding, we also fused the features with weighted sum, where a weight was learned for the contribution of each modality per channel. This method is illustrated in Figure 3. Given two feature sets $A,B\in R^{C\times D\times H\times W}$, where C is the number of channels and D, H, and W are depth, height, and width, respectively, weighted-sum applies 1×1×1 convolutions to each channel in A and B. The result is a fused feature set $F\in R^{C\times D\times H\times W}$. Considering the convolutional kernel for each channel to be $K_{i}=\left[k_{1},k_{2}\right]\in R^{2\times 1\times 1\times 1}$ each channel $F_{i}\in R^{D\times H\times W}$ in F is therefore computed as:(1)$$F_{i}=k_{1}A_{i}+k_{2}B_{i}$$
where matrices $A_{i}$ and $B_{i}$ are the *i*-th channel of the feature sets A and B. Using 1 × 1 × 1 convolutions instead of concatenation significantly reduces the number of parameters and, consequently, the computational power and amount of data needed for training the model.

#### 2.2.2. Network Training

The input bounding boxes have different sizes and the thrombus can be in different locations within the bounding box. In order to make the network invariant to these changes, we prepared the training data as follows: we created five bounding boxes for each patient in our train set. For each bounding box, we first made a tight box around the thrombus and subsequently chose a random offset between (−0.1 × thrombus size, n × maximum bounding box size) to add to the bounding box size in each direction. Five different values were chosen for n [0.2,0.3,0.5,0.7,0.9].

All networks were trained for 150 epochs with a batch size of 1 using the Adam optimizer. Since the number of thrombus voxels was smaller than the number of background voxels, there was a class imbalance between the positive and negative classes. We used weighted cross entropy loss to account for the class imbalance problem. For each network, the optimum positive weight of weighted cross entropy and initial optimizer learning rate were chosen by hyperparameter tuning on the validation set. Hyperparameter tuning was conducted using the hyperparameter sweep feature of the weights and biases package. Hyperparameter tuning was run for the same amount of time for each model. The search intervals of hyperparameter tuning were (0.0001–0.1) and (10–50) for the initial learning rate and the weight of the weighted cross entropy loss, respectively.

#### 2.2.3. Dynamic Bounding Box Algorithm

An initial 3D bounding box was automatically generated around the detected thrombus point location. Its size was pre-defined, such that the initial bounding box for all patients had the same proportions. These proportions were chosen based on the train set, such that 90% of the thrombi in the train set completely fit inside it: in the longitudinal axis, the thrombus point location was 15 and 65 voxels away from the cranial and caudal sides of the initial bounding box, respectively. In the frontal and sagittal axes, the thrombus point location was 24 voxels away from both sides of the initial bounding box.

Due to variability in size and orientation of thrombi, and also in the position of the predicted thrombus point location, the initial bounding box occasionally only partially contains the thrombus. Furthermore, in case thrombus detection is not accurate, the initial bounding box may not contain the thrombus at all. Therefore, we propose the dynamic bounding box algorithm to customize the location and proportions of the initial bounding box. The dynamic bounding box algorithm consists of the flexible bounding box and the moving bounding box methods. The flow chart of the dynamic bounding box algorithm is depicted in Figure 4. The hyperparameters of this algorithm are chosen experimentally based on the validation set. We evaluated the effectiveness of this algorithm’s methods with an ablation study based on the results of the best performing segmentation network.

##### Moving Bounding Box Method

In cases where the network’s segmentation prediction is small, there is a chance that the segmentation is erroneous because the bounding box does not contain the thrombus. Therefore, if the segmentation network’s prediction was smaller than 100 voxels, we used the moving bounding box method to explore the area around the initial bounding box. This method moved the initial bounding box $\frac{2}{3}$ of its length in 6 cube side directions to create 6 alternative bounding box candidates. Subsequently, the segmentation network was used to make a segmentation prediction for each of these candidates. Finally, the candidate bounding box that yielded the largest thrombus segmentation volume was chosen.

##### Flexible Bounding Box Method

If the predicted thrombus is close to the edge of the bounding box (5 voxels or less), there is a chance that part of the thrombus is left outside of the bounding box. In this case, the flexible bounding box method increases the size of the bounding box in that dimension by 10 voxels, and proceeds to pass the new bounding box through the segmentation network. If the resulting segmentation is still close to the edge, this procedure can be repeated up to 5 times.

#### 2.2.4. Post-Processing

Predicted segmentations in the validation set commonly included small artefacts at the edges of the large main segmentation prediction volume. Therefore, we performed morphological closing with a kernel size of 3×3×3 to connect these components to the main volume. Additionally, we omitted components that were smaller than 10 voxels.

### 2.3. Evaluation Metrics

We evaluated the segmentation results using five spatial measures: Dice, surface Dice, 95th percentile Hausdorff distance (HD), number of non-overlapping connected components, and percentage of missed thrombi; and two volumetric measures: the intraclass correlation coefficient (ICC) and Bland–Altman analysis. The use of these metrics is briefly explained in this section. Additionally, we report precision and recall for the best performing network.

#### 2.3.1. Dice

Considering $V_{p}$ and $V_{gt}$ to be the set of voxels in the automated segmentation and ground truth respectively, the Dice coefficient [26] is defined as:(2)$$Dice=\frac{2|V_{gt}⋂V_{p}|}{|V_{gt}\left|+\right|V_{p}|}$$

It should be noted that Dice is also equal to the harmonic mean of voxel-wise precision and recall. In order to calculate metrics such as precision and recall for segmentations, each voxel in the scan is defined as true/false positive/negative. Namely, a true positive is a voxel that is correctly classified as thrombus and a true negative a voxel that is correctly classified as non-thrombus. However, these metrics are sensitive to class imbalance, and are affected more severely by errors in small segments than in large segments [27]. For instance, completely missing a small object will result in a few false positives and many true negatives.

#### 2.3.2. Surface Dice

The Dice coefficient does not distinguish between the voxels on the surface and interior of the volume. As a result, a negligibly small deviation from the surface in many locations affects Dice similarly to a great deviation in one location. Nikolov et al. [28] propose surface Dice coefficient to resolve this issue. Similar to Dice, this metric is also a score between 0 and 1. Considering $S_{overlap}^{\tau }$ to be the set of points on the prediction surface that overlap with the ground truth at a specified tolerance τ (we choose τ = 1 mm) and $S_{gt}$ and $S_{p}$ to be the set of points on the surface of the ground truth and the surface of the segmentation prediction, surface Dice is defined as:(3)$$SurfaceDice=\frac{2|S_{overlap}^{\tau }|}{|S_{gt}\left|+\right|S_{p}|}$$

Even in an adequately accurate segmentation, negligibly small deviations from the ground truth can occur on the surface. Furthermore, due to the difficulty of detecting and delineating the exact contour of a lesion, ground truth annotations are also often imperfect on the surface. Surface Dice has incorporated the tolerance τ to avoid penalizing these small deviations in segmentations.

#### 2.3.3. 95th Percentile Hausdorff Distance (HD)

The original (maximum) Hausdorff distance (HD) metric is defined as the maximum of the shortest distances between two finite point sets. However, this definition is sensitive to outliers. An alternative improved measure is therefore the 95 percentile HD, which uses the 95th percentile of the distances instead of the maximum. In cases where the network did not predict anything, the 95th percentile HD was not calculated.

#### 2.3.4. Intraclass Correlation Coefficient (ICC)

We report the ICC (two-way mixed model for absolute agreement, single measure) between the volume of the thrombus segmentation and the volume of the ground truth. We interpret values less than 0.5 as poor, values between 0.5 and 0.75 as moderate, values between 0.75 and 0.9 as good, and values higher than 0.9 as excellent reliability [29].

#### 2.3.5. Number of Non-Overlapping Connected Components

We report the average number of connected components in the segmentation output that do not overlap with the ground truth at all.

#### 2.3.6. Bland–Altman Plot

The Bland–Altman analysis [30] investigates to what extent two quantitative measures concur with each other. We created a Bland–Altman plot to investigate to what degree the predicted segmentation volume agreed with the ground truth annotation volume, for the best performing network. The vertical axis of the Bland–Altman plot shows the difference between the two volumes. The horizontal axis shows the mean of the two volumes. The mean of the values on the horizontal axis is therefore a volumetric measure of the extent to which the predictions differ from the ground truth.

#### 2.3.7. Missed Cases

We report the percentage of cases where the segmentation outputs show no overlap with the ground truth (Dice = 0).

## 3. Results

The best performing network is the weighted-sum with a surface Dice of 0.78 (95% CI 0.73–0.83) and a Dice score of 0.62 (95% CI 0.58–0.66). Average precision and recall for the weighted-sum network are 0.66 (95% CI 0.62–0.70) and 0.67 (95% CI 0.62–0.73), respectively. Evaluation metrics for all networks are reported in Table 2. The average surface Dice of the dual modality networks ranges from 0.69 (95% CI 0.63–0.76) to 0.78 (95% CI 0.73–0.83) for add and weighted-sum, respectively. Excluding CTA from the U-Net model (and thus only using NCCT) strongly reduces the surface Dice of U-Net from 0.75 (95% CI 0.69–0.80) to 0.51 (95% CI 0.43–0.58).

As a measure of model complexity, the number of trainable parameters is reported in Table 2. Including a separate encoder for the CTA images, as done for the concatenate, weighted-sum, and add networks, inevitably increases complexity. A more complex model can learn more complex patterns in the data, but it has a higher chance of over-fitting to smaller datasets. Additionally, having less trainable parameters reduces the computational costs of a model. The number of trainable parameters of add is 17.0%, weighted-sum 17.8%, and concatenate 39.9%, more than dual modality U-Net.

Based on Dice, surface Dice, and the number of missed cases, the three dual modality models all performed well, with weighted-sum achieving the highest Dice, surface Dice, and the smallest number of missed cases. The average number of non-overlapping connected components in add, weighted-sum, and concatenate is less than 1, but for U-Net, this number is 2.5 (95% CI 1.5–3.5). Using only NCCT as a single modality input results in the lowest number of non-overlapping components. Despite the high surface Dice score, weighted-sum has a lower volume ICC than add and concatenate. The paired student t-test was used to test the significance of the average Dice score difference among the weighted-sum and U-Net (*p*-value = 0.023), concatenate (*p*-value = 0.12), add (*p*-value < 0.001), and U-Net with no CTA input (*p*-value < 0.001).

Since weighted-sum has the best segmentation performance, we performed the additional analyses based on the predictions of the weighted-sum model. One example of the weighted-sum predictions with the surrounding anatomy is shown in Figure 5. Additionally, in order to provide a visual representation of the segmentation prediction in a 3D bounding box, all the axial slices of the bounding box for two patients are depicted in Appendix B.

The Bland–Altman plot of thrombus volumes based on the weighted-sum model predictions is depicted in Figure 6. This plot shows that the automated segmentations tend to overestimate the volume of small thrombi and underestimate the volume for larger thrombi. The mean difference in thrombus volumes of the prediction and ground truth is −29 mm${}^{3}$ (95% CI −267–208 mm${}^{3}$).

In order to provide further insight into the segmentation predictions of weighted-sum, we report the average Dice and surface Dice for thrombi of different occlusion locations in Table 3. We also include the number of cases of each occlusion location and the number of missed cases for each location. M2 thrombi have the lowest spatial agreement with a surface Dice of 0.66 (95% CI 0.37–0.95). The model performs similarly well for ICA-T and M1 thrombi.

The ablation study on the effectiveness of the dynamic bounding box algorithm and post-processing was performed on the weighted-sum network. The results of omitting the moving bounding box method, the flexible bounding box method, and post-processing are shown in Table 4. This table shows that the dynamic bounding box methods do not strongly affect Dice. However, the moving bounding box method decreases the number of missed cases. The applied post-processing only minimally affects surface Dice, ICC, and the number of missed cases, but it increases the average number of non-overlapping components.

In 95% of the cases, the thrombus is (at least partially) included in the initial bounding box (recall of 0.95). The entire thrombus was included in the initial bounding box with a recall of 0.52. In the cases where thrombus was only partially included, on average, 78% of the thrombus volume was inside the initial bounding box. For three out of the five cases where the initial bounding box did not contain the thrombus at all, the weighted-sum network did not make any predictions, and the predicted lesion volume in the remaining two cases was negligibly small (less than 3 mm${}^{3}$). After applying the dynamic bounding box algorithm for the weighted-sum network, recall was 0.98 and 0.76 for partial and complete thrombus inclusion. The average percentage of thrombus volume that was inside the final bounding box increased to 83%.

Two examples of weighted-sum thrombus segmentations in the test set with a high agreement with the ground truth are shown in Figure 7. We observe that, in multiple cases, the network is capable of segmenting the HAS even when the contrast difference of HAS and the background is minimal. Additionally, in cases such as Figure 7B, the network is able to distinguish hyperdense areas that mimic the HAS based on information from the CTA. Figure 8 shows two cases where the predictions of weighted-sum only partially agree with the ground truth. Since the edges of the HAS are not always clearly distinguishable from the background, in some cases, it is difficult to determine the true extent of the thrombus.

Some examples of weighted-sum predictions with high Dice (above 0.6) are shown in Figure 9. Additionally, some cases with segmentation errors or lower Dice scores are shown in Figure 10. In order to also provide visual context and enable a comparison between Dice and surface Dice scores, we included these scores for the depicted 2D bounding boxes in Figure 9 and Figure 10.

## 4. Discussion

Our proposed automated thrombus segmentation method with the weighted-sum network is able to segment occluding thrombi in CT images from acute ischemic stroke patients with a high spatial overlap with the ground truth. Including CTA in addition to NCCT strongly improved thrombus segmentation accuracy. Assigning a separate encoder for each modality and combining the features using weighted-sum results in the best segmentation performance. The introduced dynamic bounding box algorithm reduced the number of missed thrombi, but did not significantly affect the Dice score.

Only a few studies have reported fully automated thrombus segmentation methods on CT scans of patients with LVO stroke. Tolhuisen et al. [12] use patch-based CNNs over the entire scan and report results on a test set containing 58 LVO cases and 45 stroke mimic cases. Their method shows a moderate volume agreement ICC (0.49) but a poor agreement ICC for thrombus density (0.14). The poor density agreement suggests that the overlap between the segmentation results and the ground truth is suboptimal. Lucas et al. [16] used cascaded CNNs and limited the search area to the MCA and ICA regions. They used their initial segmentation results as input to a classifier network to detect the affected hemisphere and used this information to further limit the search area to one hemisphere. They reported an average Dice of 0.5. Notably, their method produced a large number of thrombus candidates with three to four connected components per patient on average. Instead of limiting the search area to the MCA and ICA region for all cases, we propose first detecting the thrombus location and then using the detection results to create a customized limited search area per case as input for the segmentation network.

Our method achieves a high spatial agreement and a high ratio of detected thrombi on a large test set. The networks were trained and tested on two separate multi-center, multi-scanner datasets and, therefore, were not limited to a specific imaging protocol. Furthermore, our method is fully automated and is therefore able to produce segmentation results quickly and with minimal effort, which is of critical importance in the clinical workflow of acute stroke patients.

Having two encoders results in better spatial overlap in the cases of concatenate and weighted-sum, but not in add. Even though, in theory, in add, the network should be able to weigh each feature in the convolutional layers of the encoder before adding them, introducing the additional inductive bias of learning the weights separately results in a more effective feature fusion. Additionally, we observe that, despite having fewer learnable parameters, weighted-sum performs better than concatenate.

The dynamic bounding box algorithm methods both increase the search area for difficult cases by extending the initial bounding box or by shifting it. As a result, adding these methods reduces the number of missed cases. However, this algorithm does not have a noticeable effect on the model’s ability to segment the full extent of the thrombus.

A limitation of our method is that it depends on a third-party software for thrombus detection, but since the detection and segmentation steps are separate, StrokeViewer LVO can be replaced by other thrombus detection methods with minimal effort after tuning the initial bounding box hyperparameters.

Since the thrombus analysis pipeline relies on StrokeViewer for thrombus detection, the sensitivity and specificity of thrombus detection is not addressed in this work. Specifically, we only include cases with an LVO detected by StrokeViewer in our datasets. The sensitivity and specificity of StrokeViewer was previously assessed by Luijten et al. [31]. StrokeViewer showed a sensitivity ranging between 72% and 89%. It was able to detect LVOs with a specificity of 78%. Moreover, the number of M2 occlusions was lower than ICA-T and M1 occlusions in our datasets and, therefore, our method is biased towards patients with ICA-T and M1 occlusions.

Another limitation involves the exclusion of cases with no visible HAS from our test set, which constituted only 4% of the patients (7 out of 176). Having NCCT scans with thick slices, scans with a low-dose CT protocol, or thrombi with a low percentage of RBCs can occasionally result in a complete lack of HAS. In these cases, if a CTA scan was available, it was possible to determine the proximal side of the thrombus, but it may not have been possible to reliably determine the location of the distal side of the thrombus or its exact extent. We therefore chose to exclude these cases from our dataset due to the difficulty of providing reliable ground truth annotations. Another limitation in our annotations is caused by the difficulty of distinguishing real thrombi from other phenomena causing hyper dense vessel regions on NCCT scans, such as stagnant blood or atherosclerosis. Additionally, multiple occlusions or thrombi which partially block the artery, are difficult to distinguish from inhomogeneous thrombi that are partially highly pervious. Furthermore, in cases where the HAS has a low contrast with the background, it is difficult to reliably annotate the exact extent of the thrombus.

A third limitation of our study is the difficulty in achieving a high Dice for small object segmentation [32]. Even annotations that are considered accurate may disagree with the ground truth slightly on the surface of the volume. Since for small objects the surface to interior ratio is much higher than large objects, achieving a high Dice in small object segmentation is more difficult. Therefore, even though Dice is a more conventional method of measuring spatial overlap, surface Dice provides a better assessment of the model’s segmentations. The large difference between the Dice (0.62) and surface Dice (0.78) scores of our method indicates that Dice is negatively affected by negligible disagreements with the ground truth on the surface of the segmentation. In order to provide a means for comparison, Dice scores among four observers who performed manual thrombus annotations are reported in Appendix A.

The results of this study show that the current method is able to successfully segment the thrombus with a good overlap with the ground truth. However, additional research is needed to calculate thrombus imaging characteristics based on the automatic segmentations and study their relationship with outcome to assess the clinical utility. Furthermore, the current study should be considered as the proof of concept of the proposed method. Performance can potentially be improved by experimenting with alternatives to U-Net and incorporating methods, such as residual units, deep supervision, and dilated convolutions in follow-up studies.

Previous studies have shown a relationship between thrombus imaging characteristics calculated from sample-based annotations and stroke treatment [3,33,34,35] and functional outcome [1,2,3]. Similarly, thrombus characteristics calculated from full thrombus annotations are shown to be associated with treatment and functional outcomes [36,37], and were used to predict early recanalization [4] and the number of passes required for successful recanalization with different mechanical thrombectomy devices [5]. The association between thrombus characteristics and stroke outcome is further discussed in [38,39].

Our fully automated segmentation method can facilitate the quick, effortless, and unbiased assessment of thrombus imaging markers in the acute setting of stroke. Consequently, it can assist radiologists in making decisions about the optimal treatment strategy or thrombectomy device for stroke patients. Furthermore, an automatic thrombus segmentation method can be used to quickly and consistently annotate large clinical datasets, thereby accelerating stroke research. Implementation of our method would be possible in clinical practice and research in the short-term, in order to further evaluate its real-world results on a large scale.

## 5. Conclusions

We present a CNN-based automated method to segment occluding thrombi in baseline NCCT and CTA scans of patients with acute ischemic stroke due to a large vessel occlusion of the anterior circulation. The automated segmentations show a significant overlap with the manually annotated ground truth. Therefore, the proposed automated thrombus segmentation method can be used for robust assessment of thrombus imaging characteristics, which can be useful for both stroke research and potentially support stroke physicians in selecting the optimal stroke treatment in clinical practice.

## Figures and Tables

**Figure 1 diagnostics-12-00698-f001:**
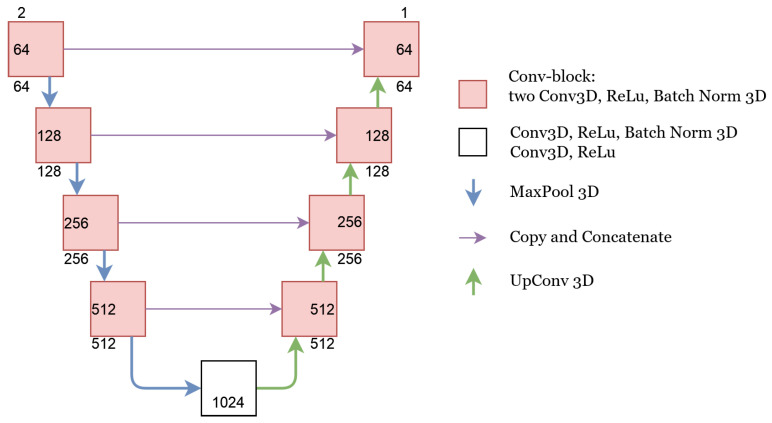
U-Net architecture. The number of input and output channels to each conv-block is depicted on the outside of the conv-block box. The number inside the conv-block box shows the number of channels after the first Conv3D operation.

**Figure 2 diagnostics-12-00698-f002:**
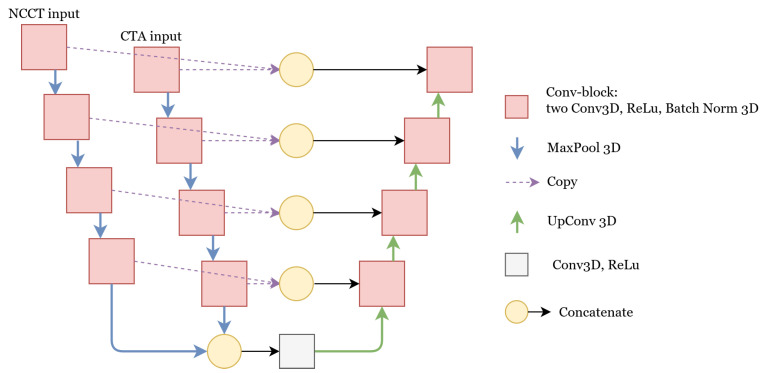
U-Net with two encoders where the features are combined using concatenation. The details of the Conv-block, MaxPool, and UpConv operations are similar to the U-Net with one encoder. For other feature fusion methods, namely add and weighted-sum, the concatenate operation was replaced by addition and weighted-sum, respectively.

**Figure 3 diagnostics-12-00698-f003:**
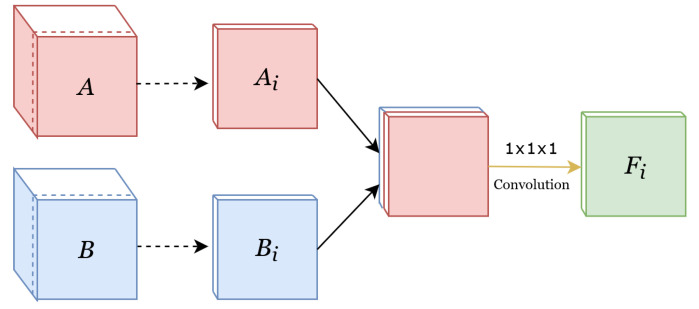
Implementation of the weighted-sum feature fusion. Since it is not possible to depict the original 4D structure, the features are depicted in 3D. A and B are the feature sets of the first and second modalities, respectively. The dotted arrows show the separation of the *i*-th channel from both the red and the blue feature-set, where $A_{i}$ and $B_{i}$ are the *i*-th channel of A and B. A convolutional operator is then used to combine these two channels into one channel, depicted as $F_{i}$.

**Figure 4 diagnostics-12-00698-f004:**
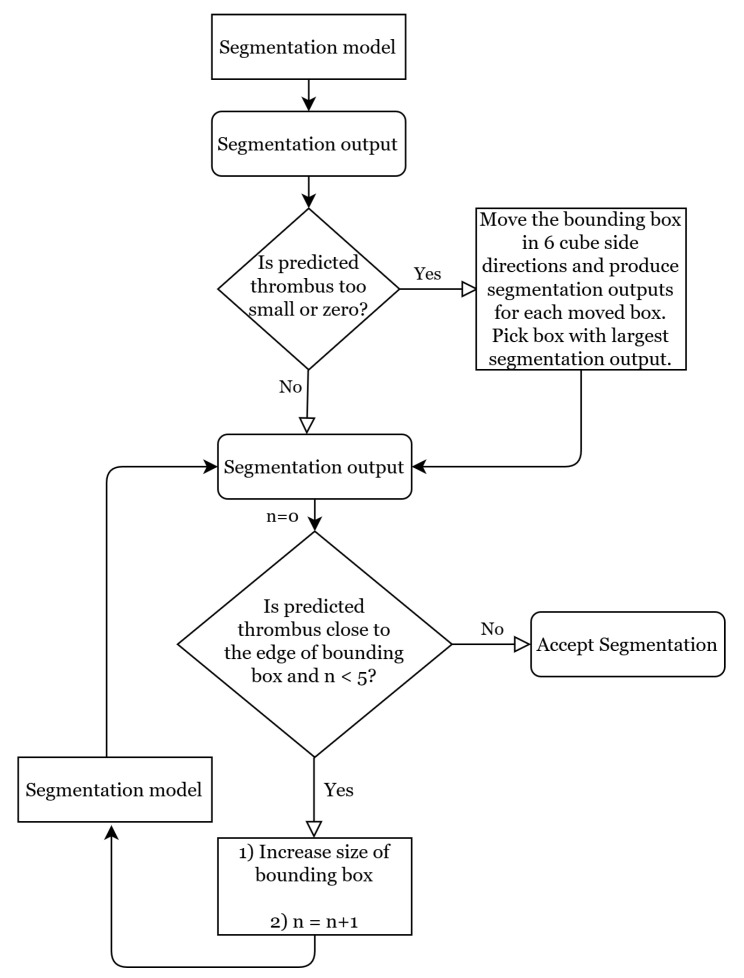
Flow chart of the dynamic bounding box algorithm.

**Figure 5 diagnostics-12-00698-f005:**
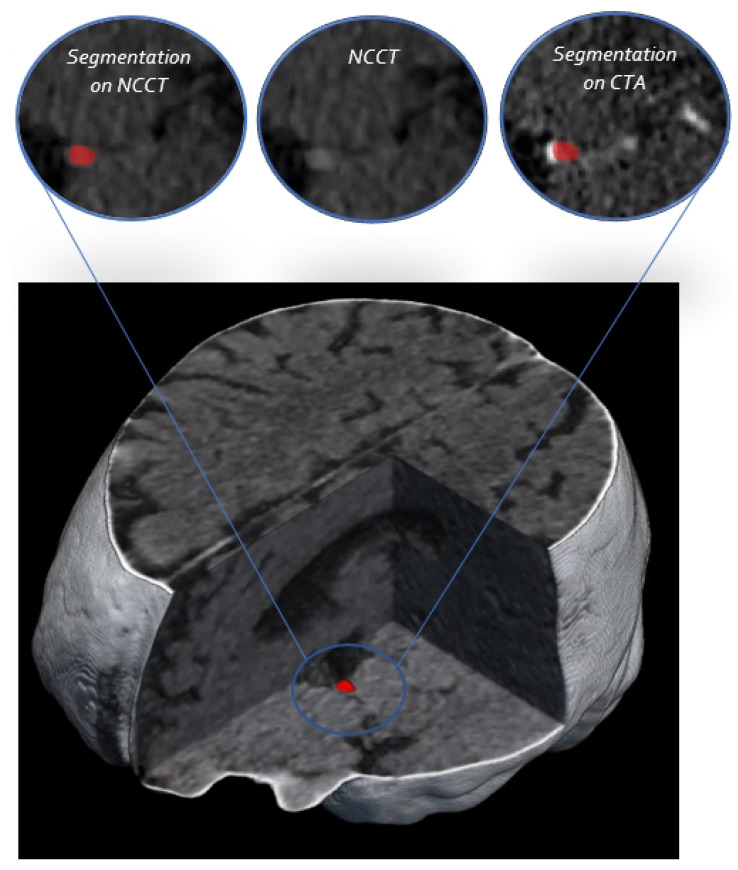
The 3D visualization of the NCCT scan. The segmentation prediction of the weighted-sum network is shown on a 2D slice of NCCT and CTA on top.

**Figure 6 diagnostics-12-00698-f006:**
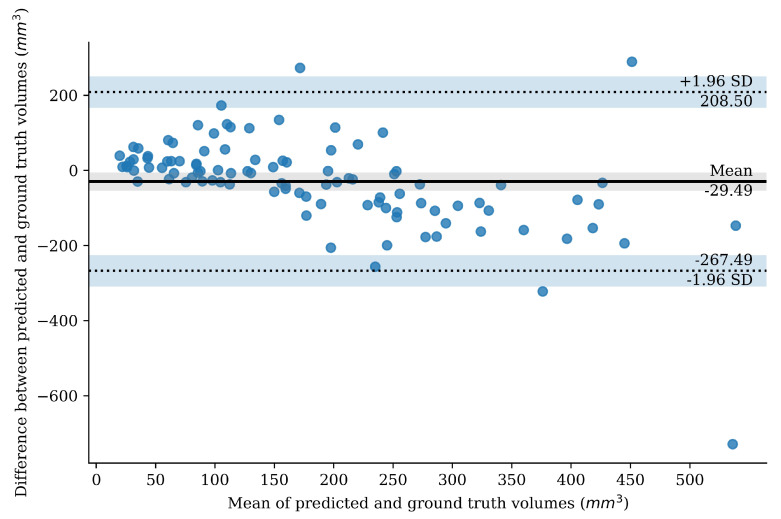
Bland–Altman plot for thrombus volume based on weighted-sum predictions. The mean volume difference is shown as a horizontal line. The 95% of the differences between the ground truth and the predictions fall within the limits of agreement that is also depicted on the plot.

**Figure 7 diagnostics-12-00698-f007:**
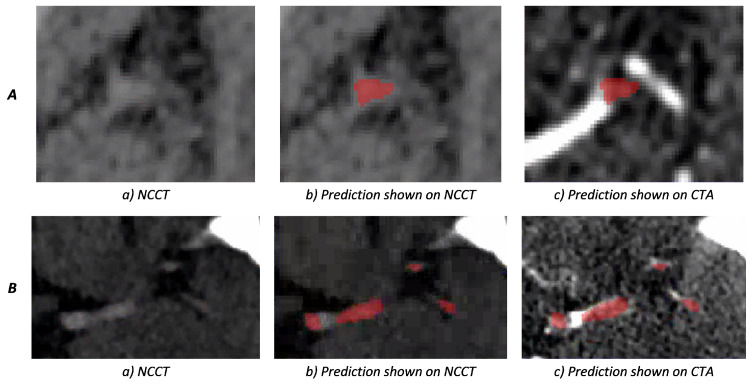
Weighted-sum thrombus segmentation results for two patients. The segmentation prediction is depicted in red; (**A**) shows a difficult case in which the network is able to successfully segment a small thrombus where the hyperdense artery sign is not easily distinguishable from the background; (**B**) depicts a case where CTA shows that the thrombus is only partially present in the artery. Part of the non-occluded artery also shows as a hyperdense region on NCCT. Therefore, information from both NCCT and CTA is required to accurately annotate the thrombus. This figure shows that the network is able to discard the area with the visible contrast on CTA from the segmentation.

**Figure 8 diagnostics-12-00698-f008:**
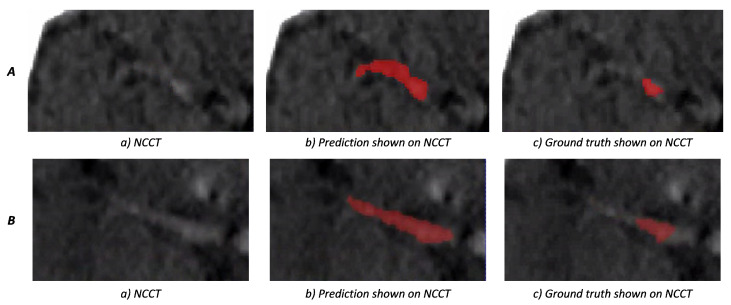
**A** and **B** both show cases where weighted-sum predictions only partially agree with the ground truth. In figures (**b**,**c**), the prediction and ground truth are shown in red, respectively. In cases where the thrombus intensity decays gradually, it can be difficult to determine the extent of the thrombus, manually and automatically.

**Figure 9 diagnostics-12-00698-f009:**
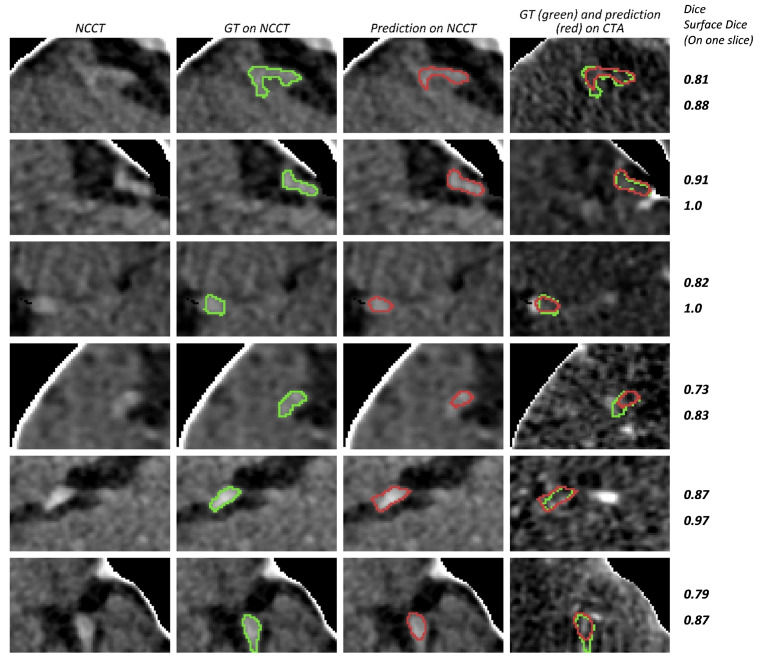
Examples of cases with a Dice score higher than 0.6. The bounding box is used to limit the scan area for each case. Since the bounding box is three-dimensional, only the axial slice with the largest ground truth surface area is depicted. Different cases are represented in rows. The first three columns show NCCT and the right most image column shows the CTA image. The boundaries of the ground truth annotation (GT) and segmentation prediction (prediction) are shown in green and red, respectively. Dice and surface Dice values for the displayed 2D slice are shown at the right side for each row.

**Figure 10 diagnostics-12-00698-f010:**
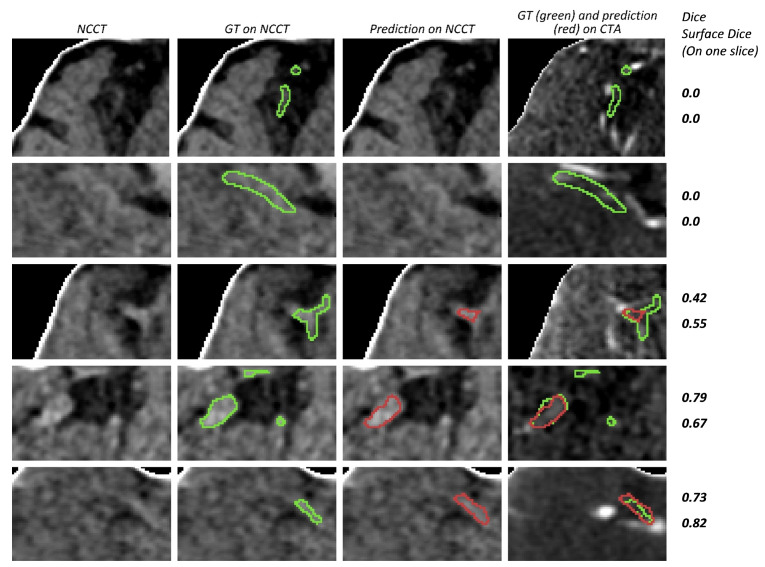
Examples of cases with segmentation errors or a low Dice score. The bounding box is used to limit the scan area. Since the bounding box is three-dimensional, only the axial slice with the largest ground truth surface area is depicted. The boundaries of the ground truth annotation (GT) and segmentation prediction (prediction) are shown in green and red, respectively. Dice and surface Dice for the displayed 2D slice are shown at the right side for each row. The top-most row shows a missed case (Dice = 0). The second row shows a case with a non-zero overall Dice score where the network does not segment the thrombus in the presented axial slice.

**Table 1 diagnostics-12-00698-t001:** Dataset characteristics. Numbers in the parenthesis show the 95% confidence interval.

Dataset	% Right Hemisphere Affected	Average Thrombus Volume (mm${}^{3}$)	Average Thrombus Length (mm)	Average Hounsfield Value of Thrombus on NCCT
Training set	55	221 (196, 246)	31 (28, 34)	47 (45, 49)
Validation set	55	187 (111, 263)	23 (16, 29)	45 (42, 48)
Test set	48	168 (147, 188)	32 (29, 36)	50 (49, 51)

**Table 2 diagnostics-12-00698-t002:** Comparing the properties and performance of different segmentation networks. Numbers in the parenthesis show the 95% confidence interval. Best performance in each metric is emphasized by bold text. Missed cases shows the percentage of cases with a Dice = 0 in the test set.

Model	Number of Trainable Parameters	Dice	Surface Dice	Number of Non-Overlapping Connected Components	Volume ICC	95th Percentile HD	Missed Cases
U-Net	82M	0.59 (0.55, 0.64)	0.75 (0.69, 0.80)	2.5 (1.5, 3.5)	**0.70**	10.46 (7.33, 13.58)	6%
Concatenate	115M	0.60 (0.55, 0.65)	0.76 (0.71, 0.82)	0.6 (0.3, 0.9)	0.68	5.89 (4.36, 7.42)	8%
Weighted-sum	97M	**0.62** (0.58, 0.66)	**0.78** (0.73, 0.83)	0.7 (0.4, 0.9)	0.60	5.88 (4.54, 7.22)	**4%**
Add	96M	0.53 (0.48, 0.58)	0.69 (0.63, 0.76)	0.3 (0.1, 0.4)	0.49	**5.67** (4.30, 7.04)	11%
U-Net with no CTA input	82M	0.38 (0.32, 0.45)	0.51 (0.43, 0.58)	**0.2** (0.1, 0.3)	0.41	7.75 (5.25, 10.26)	26%

**Table 3 diagnostics-12-00698-t003:** Average Dice and surface Dice of weighted-sum per occlusion location. Numbers in the parenthesis show 95% confidence interval. Number of thrombi for each occlusion location in the test set is shown as ‘number of cases’ and the number of cases for each occlusion location with Dice = 0 is shown as ‘missed cases’.

Metric	ICA-T	M1	M2
Dice	0.64 (0.57, 0.72)	0.63 (0.58, 0.68)	0.51 (0.29, 0.74)
Surface Dice	0.78 (0.69, 0.87)	0.79 (0.74, 0.85)	0.66 (0.37, 0.95)
$\frac{\mathrm{missed}\;\mathrm{cases}}{\mathrm{number}\;\mathrm{of}\;\mathrm{cases}}$	$\frac{0}{22}$	$\frac{3}{70}$	$\frac{1}{8}$

**Table 4 diagnostics-12-00698-t004:** Ablation study on the weighted-sum network. Numbers in the parenthesis show 95% confidence interval.

Experiment	Dice	Surface Dice	Number of Non-Overlapping Connected Components	Volume ICC	95th Percentile HD	Missed Cases
No moving bounding box	0.60 (0.55, 0.64)	0.75 (0.69, 0.80)	0.6 (0.4, 0.9)	0.67	5.58 (4.30, 6.85)	9%
No flexible bounding box	0.61 (0.57, 0.66)	0.77 (0.73, 0.82)	0.33 (0.2, 0.4)	0.63	6.39 (4.30, 8.48)	4%
No post-processing	0.62 (0.58, 0.66)	0.78 (0.73, 0.83)	0.9 (0.6, 1.3)	0.60	5.85 (4.51, 7.19)	4%

## Data Availability

The data presented in this study are available upon reasonable request from the corresponding author. The data are not publicly available due to ethical restrictions that prevent the sharing of patient data.

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
