# Peer review of "Fully Automated Thrombus Segmentation on CT Images of Patients with Acute Ischemic Stroke"

_diagnostics, 2022, doi:10.3390/diagnostics12030698_

Round 1
Reviewer 1 Report
The paper demonstrates the effectiveness of U-net based segmentation algorithms in segmenting thrombus in NCCT and CTA images. The method has several limitations including its reliance on a third-party software for thrombus detection, the exclusion of no visible HAS, and poor agreement among some experts' annotations, as was explained by the authors.
Comments are:
- The authors need to clarify 1) if the identification of the bounding box is manually done by a user via the software interface, and 2) whether the bounding box is 2-D or 3-D. A figure that shows the process of identifying a bounding box surrounding the clot on a user interface will be very helpful.
- It will be helpful to include a figure that shows slice-by-slice annotations of the thrombus in a patient's volume CT data.
- Table 2 suggests that Surface Dice is always higher than Dice. Please explain why. It intuitively makes more sense that Surface Dice should produce a lower value than Dice.
- In pages 5-6, the authors describe the methods such as Concatenate, Add, and Weighted-sum. These methods don't seem sufficient in detail, so it will be clearer if the authors provide any references that show similar methodologies, model architectures.
- In 2.1.2. Pre-processing, please explain what software tools are used to perform skull stripping and registration to an atlas brain. What atlas template was used?
- A link to source code will be very helpful for reproducible research. Please indicate its availability.
Reviewer 2 Report
The paper is well written, the flow charts, images and tables are sufficient. Maybe a paragraph in the discussion section, describing how the method of automatic thrombus segmentation could potentially affect the clinical practice (for example decide timely the appropriate thrombectomy technique or device etc), could be of further interest for the clinicians and researchers.
